# Cardiopulmonary Exercise Physiology in AL Amyloidosis Patients with Cardiac Involvement and Its Association with Cardiac Imaging Parameters

**DOI:** 10.3390/jcm11185437

**Published:** 2022-09-16

**Authors:** Alexandros Briasoulis, Foteini Theodorakakou, Athanasios Rempakos, Ioannis Petropoulos, Maria Gavriatopoulou, Emmanuel Androulakis, Kimon Stamatelopoulos, Anastasios Kallianos, Georgia Trakada, Meletios Athanasios Dimopoulos, Efstathios Kastritis

**Affiliations:** 1Department of Therapeutics, Faculty of Medicine, National and Kapodistrian University of Athens, 157 72 Athens, Greece; 2Royal Brompton, Department of Cardiology, National Healthcare System, London SW3 6NP, UK

**Keywords:** amyloidosis, cardiopulmonary exercise testing

## Abstract

Background: Cardiopulmonary exercise testing (CPET) has been widely used for the functional evaluation of patients with heart failure. Patients with amyloidosis and cardiac involvement typically present with heart failure with preserved or mildly reduced ejection fraction. We sought to evaluate the use of CPET parameters in patients with AL amyloidosis for the assessment of disease severity and prognosis and their association with cardiac imaging findings. Methods: A single-center prospective analysis was conducted, which included 23 consecutive ambulatory patients with AL amyloidosis with cardiac involvement, not requiring hospitalization or intravenous diuretics. Patient evaluation included CPET, laboratory testing, echocardiography and cardiac MRI. The cohort was divided according to the presence of high-risk CPET characteristics (below median peak VO2 and above median VE/VCO2). Results: Patients with AL amyloidosis and cardiac involvement (median age was 60 years (56.5% males) had median peak relative VO2 (VO2/kg) of 17.8 mL/kg/min, VE/VCO2 slope of 39.4 and circulatory power of 2362.5 mmHg⋅mL/kg/min. Peak relative VO2 gradually declined across Mayo stages (*p* = 0.046) and exhibited a significant inverse correlation with NT-proBNP levels (r = −0.52, *p* = 0.01). Among imaging parameters, peak VO2 positively correlated with global work efficiency (r = 0.61, *p* < 0.001), and global work index (r = 0.45, *p* = 0.04). The group of patients with high-risk CPET findings showed evidence of more advanced disease, such as higher NT-proBNP levels (*p* = 0.007), increased septal and posterior left ventricular wall thickness (*p* = 0.043 and *p* = 0.033 respectively) and decreased global work efficiency (*p* = 0.027) without substantial differences in cardiac MRI parameters. In this group of patients, peak VO2 and VE/VCO2 were not associated significantly with overall survival and cardiac response at one year. Conclusion: In patients with AL amyloidosis, evaluation of exercise capacity with CPET identified a group of patients with more advanced cardiac involvement. The potential of CPET as a risk stratification tool in AL amyloidosis with cardiac involvement warrants further research.

## 1. Introduction

Immunoglobulin light chain (AL) amyloidosis is the most common form of systemic amyloidosis, with an incidence of approximately 9 to 14 cases per million person-years [1]. AL amyloidosis is usually caused by a small, slowly proliferating bone marrow plasma cell clone, which secretes an unstable immunoglobulin light chain forming amyloid fibrils that then are deposited in the extracellular space of tissue [2]. With the exception of the central nervous system, all other organs can be affected by AL amyloidosis, and cardiac involvement is the most common and the one defining prognosis [3]. Typically, cardiac involvement in AL amyloidosis initially presents as a restrictive cardiomyopathy that progresses to heart failure with preserved or mildly reduced ejection fraction (HFpEF) [4].

Cardiopulmonary exercise testing (CPET) is a test used to assess a patient’s cardiorespiratory capacity. CPET has become an important tool for patients with heart failure (HF), since it aids in classifying disease and determining prognosis [5]. Furthermore, peak VO2 values measured during CPET have an important role in assisting patient selection for advanced HF interventions, including heart transplantation and ventricular assist devices [6]. CPET is not yet being routinely used in patients with cardiac amyloidosis. Nevertheless, it is an interesting prospect and, recently, studies assessing its value for AL amyloidosis patients have emerged [7,8], although data are still scarce. The rationale behind utilization of cardiopulmonary exercise parameters in cardiac amyloidosis patients is that an important component of functional status limitation is due to cardiac involvement leading to heart failure. Targeted therapies toward the etiology of amyloidosis prevent deterioration of disease but exert minimal effects on the echocardiographic characteristics of the left and right ventricles. However, many patients treated with disease modifying therapies and supportive care aiming at volume control experience improvement in functional status. CPET could accurately assess the functional status and the severity of heart failure at baseline before treatment, contribute to prognosis and provide valuable information on the disease course and response to treatment. The available data demonstrate severely reduced functional status on CPET and their population consists mainly of transthyretin patients. In light of the limited currently available evidence focusing on AL amyloidosis, we sought to examine cardiopulmonary exercise testing performance of AL amyloidosis patients with cardiac involvement and identify clinical, biochemical and imaging markers associated with exercise capacity impairment in this challenging patient population. 

## 2. Methods

### 2.1. Study Design

This is a single-center prospective analysis of twenty-three consecutive patients with AL amyloidosis who were ambulatory, not requiring hospitalization or intravenous diuretics in the previous month. All patients were diagnosed and underwent CPET between March 2021 and March 2022 with complete follow-up until July 2022 at the Department of Clinical Therapeutics of National Kapodistrian University of Athens (Alexandra General Hospital). We excluded newly diagnosed patients with decompensated heart failure requiring hospitalization for intravenous diuretics, inotropes and vasopressors. Also, patients wheelchaired or bed-bound were not candidates for any exercise protocol and were excluded. 

The diagnosis of light-chain cardiac amyloidosis [AL CA] was based on a combination of typical features on echocardiography, cardiac magnetic resonance imaging and histologically proven systemic AL amyloidosis according to current international recommendation [9]. Patients were diagnosed, treated and followed prospectively in our center by a group of Oncologists and Cardiologists specialized in CA. Detailed history, medication reconciliation, symptoms’ assessment, adherence, vital signs and screening for hypotension, orthostasis, symptomatic bradycardia and rhythm disturbances were performed at each visit. AL patients were classified into groups of disease prognosis and therapeutic response by using the staging system of Mayo Clinic 2004 based on combinations of NT-proBNP and cardiac TnT at presentation of diagnosis: stage I (N-terminal-pro brain natriuretic peptide B < 332 ng/L and high-sensitivity cardiac Troponin T ≥ 50 ng/L), stage II (N-terminal- pro brain natriuretic peptide B > 332 ng/L or high-sensitivity cardiac Troponin T ≥ 50 ng/L), and stage III (N-terminal- pro brain natriuretic peptide B > 332 ng/L and high-sensitivity cardiac Troponin T ≥ 50 ng/L). Stage IIIB was defined as N-terminal- pro brain natriuretic peptide B > 8500 ng/L and high sensitivity cardiac Troponin T ≥ 50 ng/L [10,11]. All patients received therapy for AL CA which involved the administration of anticlonal chemotherapy based on bortezomib. 

Demographic, clinical follow-up and laboratory data were obtained by review of the patients’ medical records and from the prospectively collected clinical database. All patients underwent laboratory testing for measurement of disease-specific biomarkers (dFLC: difference between involved and uninvolved serum free light chains, NT-proBNP and high-sensitivity cardiac troponin T), echocardiographic and myocardial work tests and MRI assessment. 

The study was approved by the Institutional Review Board and was conducted in accordance with the Declaration of Helsinki. All participants provided their informed consent to participate.

### 2.2. Cardiac MRI Acquisition Protocol and Analysis

All participants underwent a CMR examination, using a 3.0T MRI Philips (Achieva TX) manufactured scanner. CMR analysis was performed by a radiologist experienced in CMR imaging and one MRI physicist blinded to participants’ clinical history using the commercially available software (Circle cmr42 release 5.11.4; Circle Cardiovascular Imaging, Calgary, AB, Canada). Left ventricular endo- and epicardial borders were manually outlined in the short-axis slices at the end-diastolic and end-systolic phases. Left ventricular features and function, including LV wall thickness, wall mass, ejection fraction, end-systolic volume, end-diastolic volume and stroke volume were computed based on short-axis slices. Native and post contrast T1 myocardial relaxation images were firstly manually segmented, drawing endocardial and epicardial contours, and then were co-registered to eliminate motion-related artifacts, using CVI42 software. Subsequently, the automatically derived global T1, extracellular volume (ECV) and R2 maps were visually checked for the presence of artifacts. Regions of interest were drawn in artifact-free spleen area on the short-axis T1, T2 and ECV maps.

### 2.3. Echocardiography

At baseline, patients underwent a transthoracic echocardiography, including standard echocardiographic images and specific images appropriate for speckle tracking processing. Echocardiographic studies were performed by a single experienced operator using a standard commercial echocardiographic system (Vivid S70; GE Medical Systems, Milwaukee, WI, USA). Standard echocardiographic images were acquired according to the recommendations of the European and the American Associations of Echocardiography [12]. Left ventricular ejection fraction (LVEF) was estimated using the biplane method. Echocardiographic images were processed using commercially available 2D speckle tracking software (EchoPAC PC version 204; GE Medical Systems, Milwaukee, WI, USA). Speckle tracking parameters were calculated for the left ventricle (LV). Global LV longitudinal and circumferential strain (GLS and GCS respectively) were calculated from the three apical views. LV radial strain was calculated at the level of the papillary muscles. Also, the myocardial global work index (GWI; area under the curve from mitral valve closure to mitral valve opening; mm Hg%), the global constructive work [GCW] (work performed during shortening in systole adding negative work during lengthening in isovolumetric relaxation; mm Hg%) and the wasted myocardial work (negative work performed during lengthening in systole adding work performed during shortening in isovolumetric relaxation; global wasted work [GWW]; mm Hg%). The constructive work divided by the sum of constructive and wasted work provides the myocardial global work efficiency [GWE] (%) [13].

### 2.4. Cardiopulmonary Exercise Testing

CPET was performed using a stationary bicycle ergometer. Workload was determined using a ramp protocol (10 W/min), and patients exercised until exhaustion. Heart rate and blood pressure were measured during rest, throughout the exercise test, and during recovery, using a right hand cuff sphygmomanometer. Respiratory analysis was conducted using a breath-by-breath technique and an air-flow analyzer, which measured the minute ventilation (VE), respiratory rate, oxygen uptake (VO_2_), and carbon dioxide output (VCO_2_). Peak exchange values are the highest values achieved during the last minute of exercise. Anaerobic threshold was defined as the point of exercise when carbon dioxide output increased exponentially compared to oxygen consumption. Percent predicted peak VO_2_ was calculated using peak measured VO_2_ values divided by the reference peak VO_2_ values reported by Wasserman [14]. Linear regression was used to calculate the VE/VCO_2_ slope. Circulatory power was determined by multiplying the relative peak VO_2_ (VO_2_/kg) by the peak systolic blood pressure. Also, end-tidal CO2 was measured, and breathing reserve (%) was estimated as the difference between maximal voluntary ventilation and maximal ventilation at peak exercise divided by maximal voluntary ventilation.

### 2.5. Endpoints

The main crude endpoints examined were: overall survival, any organ response and cardiac response at one-year (defined as NT-proBNP reduction >30% and >300 ng/L in patients with baseline NT-proBNP ≥ 650 ng/L) [11]. 

## 3. Statistical Analysis

The Shapiro-Wilk test was applied to determine if the estimated parameters are well-modeled by a normal distribution. Continuous variables were summarized using median and inter-quartile range, while categorical variables were listed as frequencies and percentages. Independent samples t-test and ANOVA tests were performed for comparisons between groups of continuous variables, while the chi-squared test was used to compare categorical variables. Skewed data were tested with the Kruskal-Wallis test. Pearson’s correlation coefficients and Spearman’s correlation coefficients were calculated for normally distributed and skewed data, respectively, to measure the strength and direction of association that exist between CPET parameters and certain laboratory and imaging features of patients. Survival and any response (death, organ progression, hematologic progression, need for dialysis) were assessed with Cox regression analysis. Cardiac response at one year was assessed with logistic regression analysis. Receiver operating characteristic (ROC) curve analysis was used to investigate the prognostic value of CPET results and the NT-proBNP biomarker, and the area under the curve (AUC) was calculated.

## 4. Results

### 4.1. Patient Characteristics

A total of 23 patients with AL amyloidosis were included in the analysis. Baseline patient characteristics, CPET and echocardiographic and cardiac MRI parameters are presented in Table 1. The median age was 60 years, and 56.5% of patients were males. Renal involvement was present in 54.5% of the patients, nerve involvement in 26.1%, soft tissue involvement in 21.7% and liver involvement in 17.4%. At diagnosis, patients had median NT-proBNP of 2512 pg/mL (IQR: 882, 3102.5), hs-troponin T of 41.46 ng/mL (IQR: 21.32, 61.48), and eGFR of 85.67 mL/min/1.73 m^2^ (IQR: 70.13, 105.37) and dFLC of 366.6 mg/L (IQR: 87.97, 727.47). According to the Mayo stage classification, 4 patients (17.4%) were at stage I, 11 (47.8%) were at stage 2 and 8 (34.8%) were at stage 3. Regarding NYHA classification at diagnosis, 8 patients (34.8%) were at class I, 12 (52.2%) were at class II and 3 (13%) at class IIIa. 

### 4.2. CPET Parameters

All 23 patients underwent CPET before the institution of any anticlonal therapy. The median peak relative VO_2_ (VO_2_/kg) was 17.8 mL/kg/min (IQR: 14.9, 20.45), the median peak absolute VO_2_ was 1251 mL/min (IQR: 1068.5, 1482), and the median predicted VO_2_ was 71% (IQR: 58.45, 81.95). The VE/VCO_2_ slope was 39.4 (IQR: 35.90, 42.65) and the circulatory power was 2362.5 mmHg·mL/kg/min (IQR: 1941.50, 2796.75). The median peak systolic blood pressure, diastolic blood pressure and heart rate observed during exercise testing were 140 mmHg, 80 mmHg and 120 bpm, respectively. Metabolic equivalents (METs) were also calculated, with a median value of 5.1, while the median respiratory quotient was 1.2. Lastly, the breathing reserve of patients undergoing CPET was 50.2%, the median VO2 measurement at the anaerobic threshold was 903 mL/min, and the median end-tidal CO2 pressure (PETCO2) was 29 mmHg.

### 4.3. Cardiac Imaging Parameters

All patients underwent imaging modalities, including echocardiography and magnetic resonance imaging (MRI). Echocardiography measurements revealed a median intraventricular septum (IVS) thickness of 14 mm, a posterior wall (POW) thickness of 14 mm, a left ventricular ejection fraction (LVEF) of 55%, and a global longitudinal strain (GLS) of −15.3%. The median values of myocardial work quantification were: global work index (GWI) of 1257 mmHg% (as a reference the normal value for GWI is considered to be 1896 ± 308 mm Hg%), global constructive work (GCW) of 1859 mmHg%, global wasted work (GWW) of 114 mmHg%, and global work efficiency (GWE) of 90%. Cardiac magnetic resonance imaging (CMR) analysis demonstrated a median native T1 value of 1437.5 ms, median ECV value of 40.5%, T2 value of 56.8 and GLS of −10.47%.

### 4.4. High-Risk CPET Profile

According to the medians of VO_2_/kg and VE/VCO_2_, which were used as cutoff values, we dichotomized the patients into a group with more severely impaired exercise physiology according to peak relative VO2 and VE/VCO2. Their characteristics were compared with the remaining patients not exhibiting these derangements. Patients with both a VO2/kg ≤ 17.8 and a VE/VCO2 ≥ 39.4 were placed in group A (*n* = 8), while the rest of patients were placed in group B (*n* = 15). Medians of the patient characteristics, CPET results, echocardiography and myocardial work, as well as MRI features of the two groups were compared and are presented in Table 2.

Patients in group A had significantly higher baseline levels of NT-proBNP (3414.5 vs. 2133; *p* = 0.007) and significantly lower levels of eGFR (71.21 vs. 92.42; *p* = 0.039) compared to patients in group B. Their high-sensitivity cardiac troponin T was also higher, although the result was not statistically significant (52.15 vs. 27.04; *p* = 0.081). Several echocardiographic and myocardial work features presented also significant differences between the two groups, including IVS thickness (16 vs. 12.5; *p* = 0.043), POW thickness (16 vs. 12.5; *p* = 0.033) and GWE (86.5 vs. 93; *p* = 0.027). No significant differences were observed on MRI parameters. 

### 4.5. Correlation between Exercise Parameters and Other Measures of Cardiac Involvement

The correlation of the CPET results with Mayo stage, NT-proBNP and various imaging characteristics of the patients, including CMR results and myocardial work, was assessed. The peak relative VO2 gradually decreased in more advanced Mayo stages (*p* = 0.046), as presented in Figure 1. Furthermore, VO_2_/kg was significantly correlated with the baseline NT-proBNP levels (r = −0.52; 95% CI: −0.77, −0.13; *p* = 0.01), while the correlation between NT-proBNP and the VE/VCO_2_ slope was not significant (r = 0.37; 95% CI: −0.06, 0.69; *p* = 0.08) (Figure 2). A correlation of CMR results with the VO2/kg and the VE/VCO2 slope was investigated, but results were not statistically significant, as seen on Figure 3. Lastly, GWE (r = 0.61; 95% CI: 0.25, 0.83; *p* < 0.001) and GWI (r = 0.45; 95% CI: 0.02, 0.74; *p* = 0.04) both demonstrated significant positive correlation with the VO_2_/kg (Figure 4). Furthermore, we identified correlations between circulatory power and NT-proBNP levels (r = −0.52; 95% CI: −0.77, −0.13; *p* = 0.01) and PETCO2 (r = −0.55, 95% CI: −0.79, −0.18, *p* = 0.006).

### 4.6. CPET Prognostic Value

The prognostic value of CPET results was studied using the predefined endpoints of overall survival, any organ response and cardiac response. Cox regression was used to evaluate possible correlation between the VO_2_/kg values and overall survival or any organ response, as well as the risk of group A patients for mortality and any organ response, but the results were not significant, as seen on Table 3. Logistic regression was used to assess whether Group B patients were more likely to achieve a cardiac response at one year, but it did not produce statistically significant results (Table 4).

### 4.7. ROC Curves

ROC curves for VO_2_/kg, VE/VCO, and NT-proBNP were created and are presented in Figure 5. The AUC for the VO_2_/kg was 60.5% (95% CI: 22.9%, 98.1%), for the VE/VCO_2_ slope it was 61.2% (95% CI: 19.6%, 100%) and for the NT-proBNP it was 65.8% (95% CI: 29.9%, 100%).

## 5. Discussion

In this prospectively collected cohort of patients with AL amyloidosis who underwent CPET and comprehensive cardiac imaging assessment, we identified a gradual decline in peak VO2 in more advanced stages of disease and correlation with echocardiographic parameters, such as global work efficiency and global work index. A group of patients with severe impairment in exercise capacity characterized by below median peak relative VO2 and above median ventilatory efficiency slope had biochemical and imaging evidence of more advanced disease. In this small cohort of AL patients with cardiac involvement, we did not identify significant association between CPET performance and disease outcomes; however, the small number of patients and the limited number of events does not allow for a meaningful survival analysis.

AL amyloidosis is the most common type of amyloidosis, also having the worst prognosis. Despite that, advances have been made both in early diagnosis and treatment, resulting in an improved survival [15]. This is proof that further advances in early detection, assessment of disease severity and timely management are key elements to prolonging survival and treatment response. Special attention should be paid to early identification of cardiac involvement in AL amyloidosis patients, since this constitutes the most significant cause of mortality [4]. Biomarkers, including NT-proBNP, troponin T, and free light chains, when combined, constitute the cornerstones of disease prognostication, and response to treatment [16]. However, biomarkers, particularly NT-proBNP, can be affected by kidney function, resulting in inaccurate values in patients with renal failure, a condition common among patients with AL amyloidosis [17]. Moreover, none of these parameters provide objective information on the functional status and exercise capacity of patients with AL amyloidosis. In our study, we evaluated CPET as a tool that could potentially aid in assessing disease severity and functional status independently from other biomarkers. Furthermore, we investigated correlations between exercise impairment and other imaging aspects of AL amyloidosis.

Patients with cardiac AL classically present with HFpEF or mildly reduced ejection fraction, and CPET has been proven to be efficient at stratifying risk in patients with HFpEF [18]. Furthermore, recent studies that evaluated the use of CPET in patients with AL or transthyretin amyloidosis yielded promising results. Hein and colleagues performed CPET in 27 patients with various forms of systemic amyloidosis and showed that peak VO2 was an independent predictor of mortality among patients with cardiac involvement [7]. Similarly, Nicol and colleagues demonstrated that peak VO2 and circulatory power are independent predictors of mortality and HF hospitalization in patients with cardiac amyloidosis [8]. Lastly, in a recent analysis of 41 patients with AL or transthyretin amyloidosis, Bhutani and colleagues indicated that, peak VO2 is an indirect marker of light chain toxicity [9]. Our findings support gradual impairment of peak VO2 in more advanced stages of disease. This has important prognostic implications not only in terms of the overall survival but also regarding quality of life and exercise capacity, as with worse cardiac involvement, a phenotype of severe impairment in CPET parameters was identified.

As demonstrated by the study results, a significant correlation between CPET results and certain biomarkers, including the NT-proBNP and the eGFR of patients, was found. A significant correlation was also noted between VO2/kg and the Mayo stage of patients, which is one of the most important tools for assessing cardiac AL amyloidosis. Additionally, significant correlations were observed when comparing CPET results with echocardiographic features of patients and myocardial work results. Myocardial work is an emerging, non-invasive tool, which can be used to explore heart function and has been recently shown to yield a better predictive value than LVEF and myocardial contraction fraction in patients with cardiac amyloidosis [19]. No significant correlations were found between CPET and MRI results. This suggests that although cardiac MRI is the gold standard for early detection and diagnosis of cardiac involvement by amyloidosis, the measured parameters, such as T1 and ECV, do not play a significant prognostic role. In contrast, a functional evaluation of cardiac performance with myocardial work indices correlates with cardiac response to exercise.

Regarding the prognostic value of CPET, no correlation was found in our study between CPET and mortality. We also evaluated the data for possible correlations between CPET results and any response or cardiac response without significant results. This is contrary to the results of previous studies by Bhutani et al. and Hein et al., who demonstrated that several values derived from CPET can aid in determining prognosis [7,9]. This is probably explained by the low mortality rate (*n* = 4) in our study, as our CPET data correlate well with markers that have a well-established prognostic value, such as NT-proBNP and the Mayo Clinic Staging System. Moreover, median peak VO2 in previous studies was substantially lower compared with our measurements. This shows that our cohort included patients diagnosed in earlier stages of disease. Therefore, even in this cohort of less sick patients, peak relative VO2 correlates with disease severity.

Our study has some limitations, indicating that the results should be interpreted with caution. First, our study does not have a control group and, as such, has the fundamental limitations of its design. The number of patients was also relatively small, the follow-up was relatively short and our cohort’s low event rate could have limiting effects on survival analysis. The main strength of this analysis lies in the comprehensive evaluation of AL amyloidosis patients with cardiac imaging and CPET.

In conclusion, CPET can become an important tool in assessing disease severity in patients with cardiac AL amyloidosis, giving a wide array of information, without having the drawbacks of biomarkers. It can possibly play a pivotal role in evaluating response and candidacy for advanced cardiac therapies. Although the prognostic value of CPET was not directly demonstrated in our cohort, the CPET results correlated well with well-established prognostic markers, indicating the need for larger cohorts that could produce significant results.

## Figures and Tables

**Figure 1 jcm-11-05437-f001:**
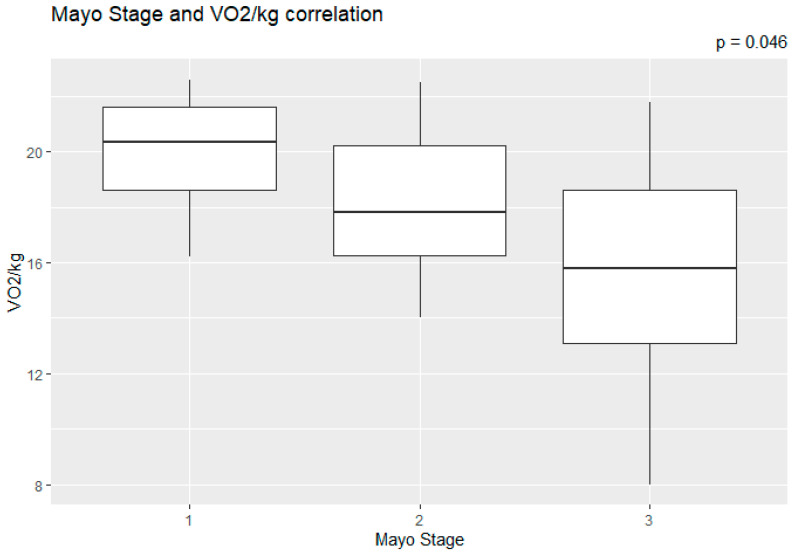
Peak relative VO2 in different Mayo stages.

**Figure 2 jcm-11-05437-f002:**
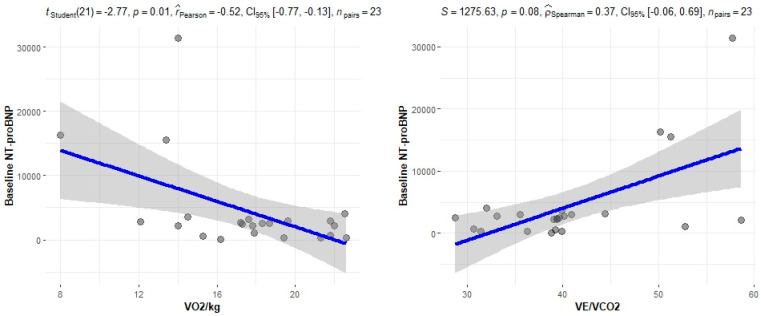
Correlation between CPET parameters and NT-proBNP.

**Figure 3 jcm-11-05437-f003:**
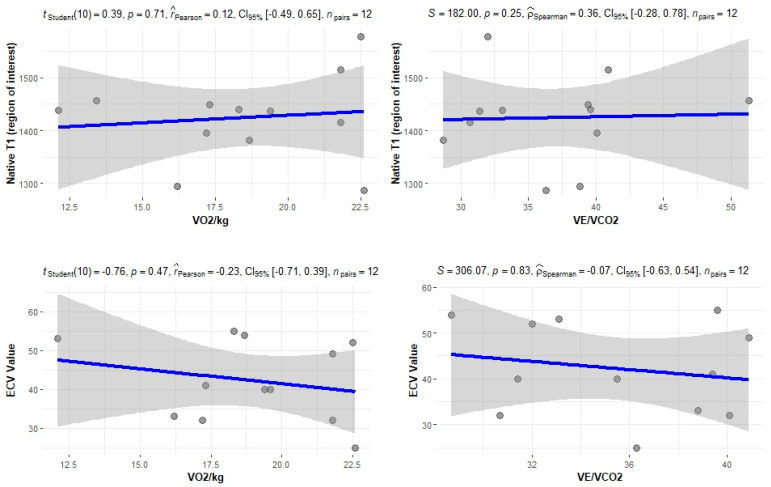
Correlation between CPET parameters and cardiac MRI findings.

**Figure 4 jcm-11-05437-f004:**
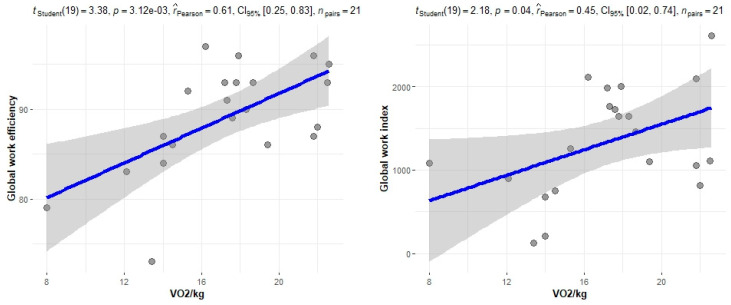
Correlation between CPET parameters and myocardial work.

**Figure 5 jcm-11-05437-f005:**
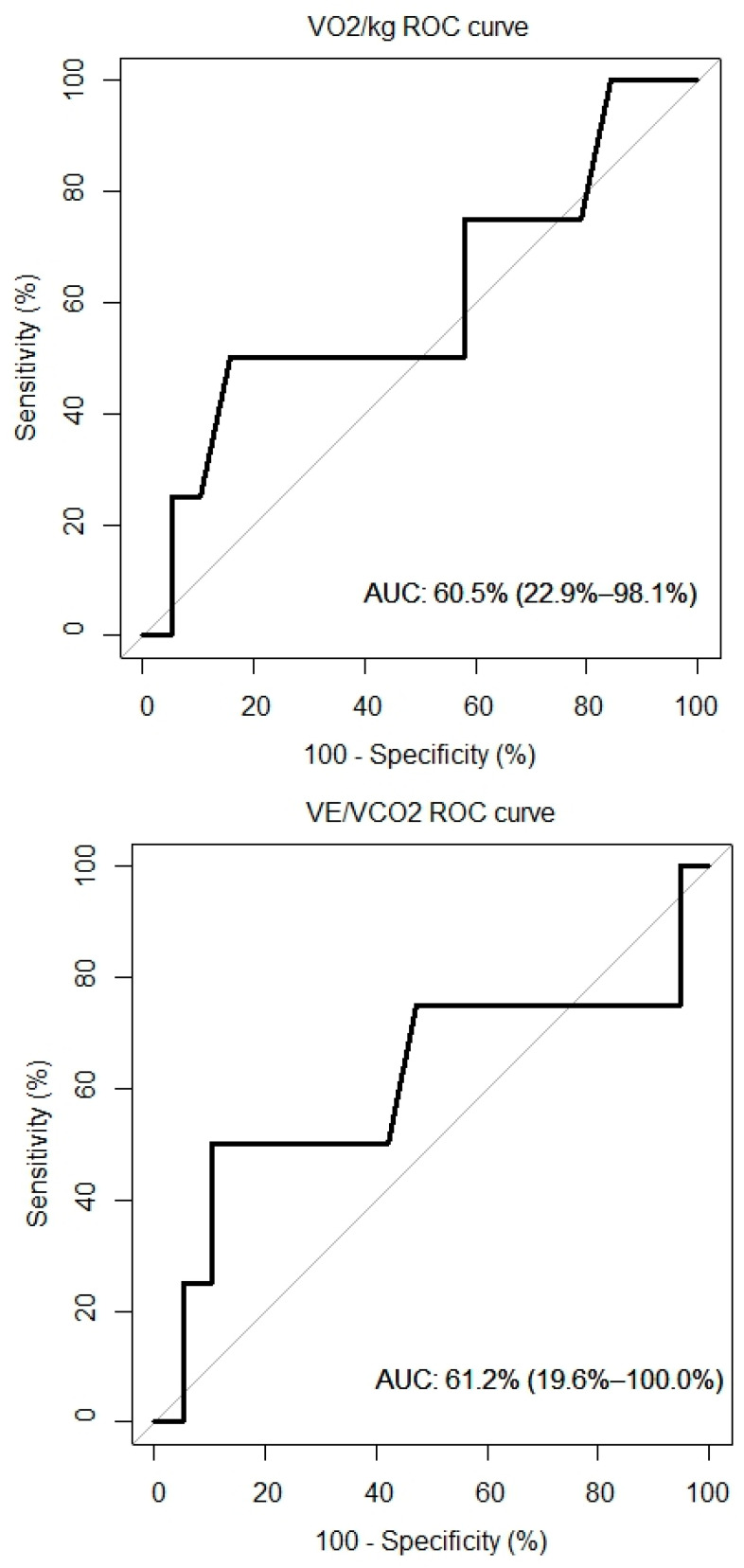
ROC curves of peak VO2, VE/VCO2 and overall survival.

**Table 1 jcm-11-05437-t001:** Baseline clinical, laboratory, exercise and imaging characteristics of patients with AL amyloidosis.

Variable	Overall (*n* = 23)
Baseline Patient Characteristics	
Age, years	60.00 [56.00, 70.00]
Gender = Male, *n* (%)	13 (56.5)
BMI, kg/m^2^	24.34 [22.65, 27.09]
Systolic Blood Pressure, mmHg	109.00 [103.50, 123.00]
Diastolic Blood Pressure, mmHg	74.00 [65.50, 79.00]
Lambda Light Chain Type, *n* (%)	17 (73.9)
Renal Involvement, *n* (%)	12 (54.5)
Liver Involvement, *n* (%)	4 (17.4)
Nerve Involvement, *n* (%)	6 (26.1)
Soft Tissue Involvement, *n* (%)	5 (21.7)
Baseline NT-proBNP, pg/mL	2512.00 [882.00, 3102.50]
High-Sensitivity Cardiac Troponin T, ng/mL	41.46 [21.32, 61.48]
eGFR, mL/min/1.73 m^2^	85.67 [70.13, 105.37]
Mayo Stage	
1, *n* (%)	4 (17.4)
2, *n* (%)	11 (47.8)
3, *n* (%)	8 (34.8)
Baseline NYHA class	
I, *n* (%)	8 (34.8)
II, *n* (%)	12 (52.2)
III, *n* (%)	3 (13.0)
Loop Diuretics, *n* (%)	11 (47.8)
Pacemaker, *n* (%)	2 (8.7)
CPET Results	
Peak Relative VO_2_, mL/kg/min	17.80 [14.90, 20.45]
Peak Absolute VO_2_, mL/min	1251.00 [1068.50, 1482.00]
Predicted Peak VO_2_, %	71.00 [58.45, 81.95]
METs	5.10 [4.30, 5.90]
RQ	1.20 [1.11, 1.24]
VE/VCO_2_ Slope	39.40 [35.90, 42.65]
Breathing Reserve, %	50.20 [38.45, 60.15]
Anaerobic Threshold VO_2_, mL/min	903.00 [769.50, 1012.50]
PetCO_2_ mmHg	29.00 [25.50, 32.00]
Peak Systolic Blood Pressure, mmHg	140.00 [132.50, 145.00]
Peak Diastolic Blood Pressure, mmHg	80.00 [80.00, 85.00]
Peak Heart Rate, bpm	120.00 [109.50, 137.50]
Circulatory Power, mmHg⋅mL/kg/min	2362.50 [1941.50, 2796.75]
Echocardiography	
Baseline IVS Thickness, mm	14.00 [12.00, 16.00]
Baseline PW Thickness, mm	14.00 [12.00, 15.50]
Baseline Mean Wall Thickness, mm	14.00 [12.00, 16.00]
Baseline LVEF, %	55.00 [49.00, 61.00]
Mitral E/Ea	17.00 [11.40, 19.75]
STDI, cm/s	12.00 [10.50, 13.50]
TAPSE, mm	19.00 [18.00, 23.25]
GLS, %	−15.30 [−18.68, −12.35]
Global Work Index, mmHg%	1257.00 [897.00, 1759.00]
Global Constructive Work, mmHg%	1859.00 [1241.00, 2084.00]
Global Wasted Work, mmHg%	114.00 [90.00, 143.00]
Global Work Efficiency, %	90.00 [86.00, 93.00]
Cardiac MRI Parameters	
Myocardial Native T1, ms	1437.50 [1392.50, 1451.00]
Myocardial ECV, %	40.50 [32.75, 52.25]
Myocardial T2 map, ms	56.80 [52.65, 58.10]
Spleen Native T1, ms	1382.00 [1344.50, 1421.00]
Spleen ECV, %	40.20 [35.80, 50.05]
Liver Native T1, ms	839.00 [803.50, 880.00]
Liver ECV, %	34.90 [30.95, 43.75]
GLS, %	−10.47 [−12.46, −9.14]

Abbreviations: BMI, body mass index; eGFR, estimated glomerular filtration rate; NT proBNP, N terminal pro brain natriuretic peptide; PETCO2, patient end tidal CO2; RQ, respiratory quotient; VO2, oxygen uptake; VE/VCO2, ventilatory efficiency slope; ECV, extracellular volume; E/Ea, ratio of peak velocity blood flow from left ventricular relaxation in early diastole to peak velocity flow in late diastole caused by atrial contraction; GLS, global longitudinal strain; STDI, S wave at tissue Doppler Imaging of the Right Ventricle; TAPSE, tricuspid annular plane excursion; IVS, interventricular septum; PW, posterior wall of the left ventricle.

**Table 2 jcm-11-05437-t002:** Characteristics of patients according to cardiopulmonary exercise parameters suggestive of severe functional impairment.

Variable	Group A (*n* = 8) (VO2/kg ≤ 17.8 & VE/VCO2 ≥ 39.4)	Group B (*n* = 15)	*p*-Value
Patient Characteristics			
Age, years	65.00 [58.75, 69.50]	57.00 [55.50, 67.50]	0.333
Gender = Male, *n* (%)	6 (75.0)	7 (46.7)	0.388
BMI, kg/m^2^	27.09 [23.97, 29.75]	23.53 [22.65, 25.05]	0.138
Systolic Blood Pressure, mmHg	105.00 [103.75, 107.25]	121.00 [103.00, 124.50]	0.245
Diastolic Blood Pressure, mmHg	74.50 [66.50, 75.00]	70.00 [65.50, 81.50]	0.722
Lambda Light Chain Type, *n* (%)	6 (75.0)	11 (73.3)	1
Renal Involvement, *n* (%)	4 (50.0)	8 (57.1)	1
Liver Involvement, *n* (%)	3 (37.5)	1 (6.7)	0.2
Nerve Involvement, *n* (%)	3 (37.5)	3 (20.0)	0.68
Soft Tissue Involvement, *n* (%)	1 (12.5)	4 (26.7)	0.8
Heart Involvement, *n* (%)	8 (100.0)	12 (80.0)	0.48
Baseline NT-proBNP, pg/mL	3414.50 [2650.00, 15708.75]	2133.00 [436.50, 2658.00]	* 0.007
High-Sensitivity Cardiac Troponin T, ng/mL	52.15 [38.02, 100.30]	27.04 [12.70, 53.34]	0.081
eGFR, mL/min/1.73 m^2^	71.21 [65.32, 84.56]	92.42 [80.74, 109.87]	0.039
Mayo Stage			0.227
1, *n* (%)	0 (0.0)	4 (26.7)	
2, *n* (%)	4 (50.0)	7 (46.7)	
3, *n* (%)	4 (50.0)	4 (26.7)	
Baseline NYHA class			0.188
1, *n* (%)	1 (12.5)	7 (46.7)	
2, *n* (%)	5 (62.5)	7 (46.7)	
3, *n* (%)	2 (25.0)	1 (6.7)	
Loop Diuretics, *n* (%)	6 (75.0)	5 (33.3)	0.142
Pacemaker, *n* (%)	1 (12.5)	1 (6.7)	1
CPET Results			
Peak Relative VO_2_, mL/kg/min	14.25 [13.85, 17.22]	19.40 [17.85, 21.80]	* 0.002
Peak Absolute VO_2_, mL/min	1188.00 [925.00, 1301.25]	1292.00 [1135.00, 1577.50]	0.197
Predicted Reak VO_2_, %	58.45 [48.88, 67.72]	74.60 [66.50, 85.95]	0.053
METs	4.10 [3.95, 4.90]	5.50 [5.10, 6.20]	* 0.002
RQ	1.21 [1.18, 1.21]	1.16 [1.09, 1.26]	0.582
VE/VCO_2_ Slope	42.25 [39.77, 50.48]	38.80 [32.55, 39.75]	* 0.018
Breathing Reserve, %	46.95 [33.70, 59.57]	54.80 [42.10, 61.45]	0.498
AT VO_2_, mL/min	826.50 [735.50, 954.00]	930.00 [877.00, 1012.50]	0.349
PetCO_2_ mmHg	25.50 [22.75, 29.25]	31.00 [27.50, 34.50]	* 0.026
Peak Systolic Blood Pressure, mmHg	137.50 [127.50, 145.00]	140.00 [137.50, 150.00]	0.327
Peak Diastolic Blood Pressure, mmHg	80.00 [80.00, 85.00]	85.00 [80.00, 90.00]	0.33
Peak Heart Rate, bpm	118.00 [100.25, 123.00]	122.00 [116.00, 151.00]	0.218
Circulatory Power, mmHg⋅mL/kg/min	2268.00 [1890.00, 2508.50]	2776.90 [2425.32, 3108.00]	0.123
Echocardiography—Myocardial Work			
Baseline IVS Thickness, mm	16.00 [14.25, 16.00]	12.50 [11.75, 14.50]	* 0.043
Baseline POW Thickness, mm	16.00 [14.00, 16.00]	12.50 [10.75, 14.25]	* 0.033
Baseline Mean Wall Thickness, mm	16.00 [14.00, 16.00]	12.50 [11.75, 14.50]	0.065
Baseline LVEF, %	55.00 [41.50, 57.50]	55.00 [50.00, 64.25]	0.308
E/Ea	19.00 [17.00, 19.50]	14.00 [10.60, 20.00]	0.388
STDI, cm/s	12.00 [11.00, 13.00]	12.50 [9.75, 14.25]	0.733
TAPSE, mm	20.00 [18.00, 23.50]	19.00 [16.50, 22.25]	0.414
GLS, %	−12.50 [−15.45, −9.90]	−16.50 [−19.95, −13.70]	0.123
Global Work Index, mmHg%	913.00 [557.75, 1735.00]	1460.00 [1104.00, 2001.00]	0.111
Global Constructive Work, mmHg%	1393.00 [792.00, 1971.25]	1882.00 [1302.00, 2231.00]	0.148
Global Wasted Work, mmHg%	125.00 [94.50, 175.50]	114.00 [76.00, 120.00]	0.346
Global Work Efficiency, %	86.50 [82.75, 89.50]	93.00 [88.00, 95.00]	* 0.027
MRI Results			
Myocardial Native T1, ms	1449.00 [1422.50, 1453.00]	1437.00 [1382.00, 1440.00]	0.518
Myocardial ECV, %	36.50 [34.25, 38.75]	44.50 [34.75, 52.75]	0.451
Myocardial T2 map, ms	56.50 [54.80, 58.65]	57.10 [51.30, 57.20]	0.926
Spleen Native T1, ms	1343.00 [1338.50, 1344.50]	1407.50 [1374.00, 1448.25]	0.066
Spleen ECV, %	42.15 [37.08, 47.22]	40.20 [37.20, 47.80]	0.814
Liver Cative T1, ms	866.00 [805.50, 894.50]	838.50 [810.25, 878.00]	1
Liver ECV, %	41.85 [38.38, 45.32]	31.40 [30.80, 40.80]	0.346
GLS, %	−9.18 [−9.82, −8.46]	−11.50 [−12.87, −9.26]	0.176

Abbreviations: BMI, body mass index; eGFR, estimated glomerular filtration rate; NT proBNP, N terminal pro brain natriuretic peptide; PETCO2, patient end tidal CO2; RQ, respiratory quotient; VO2, oxygen uptake; VE/VCO2, ventilatory efficiency slope; ECV, extracellular volume; E/Ea, ratio of peak velocity blood flow from left ventricular relaxation in early diastole to peak velocity flow in late diastole caused by atrial contraction; GLS, global longitudinal strain; STDI, S wave at tissue Doppler Imaging of the Right Ventricle; TAPSE, tricuspid annular plane excursion; IVS, interventricular septum; PW, posterior wall of the left ventricle. * denotes statistical significance.

**Table 3 jcm-11-05437-t003:** Cox regression analysis.

Regression Analysis (VO2/kg)			
Endpoint	HR	95% CI	*p*-Value
Overall survival	1.141	0.82, 1.58	0.428
Any organ response	1.008	0.78, 1.31	0.952
Regression Analysis (VO2/kg ≤ 17.8 & VE/VCO2 ≥ 39.4)			
Endpoint	HR	95% CI	*p*-Value
Mortality	0.523	0.073, 3.71	0.517
Any organ response	0.529	0.11, 2.62	0.436

VO2; oxygen uptake, VE/VCO2; ventilatory efficiency slope, HR; Hazard ratio.

**Table 4 jcm-11-05437-t004:** (**a**). Logistic regression analysis of cardiac response at one year. (**b**). Logistic regression analysis of cardiac response at one year among patients with high-risk CPET profile.

(a)			
Regression Analysis (Cardiac Response)			
Term	OR	Standard Error	*p*-Value
Intercept	0.6	0.730	0.484
VO2/kg ≥ 17.8 or VE/VCO2 ≤ 39.4	1.11	0.975	0.914
**(b)**			
**Regression Analysis (Cardiac Response)**			
Outcome	OR	95% CI	*p*-Value
VO2/kg ≤ 17.8 & VE/VCO2 ≥ 39.4 (reference)	—	—	—
VO2/kg ≥ 17.8 or VE/VCO2 ≤ 39.4	1.11	0.16, 8.07	>0.9

VO2; oxygen uptake, VE/VCO2; ventilatory efficiency slope, OR; Odds ratio.

## Data Availability

Data are available upon written request.

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
