# Peer review of "Cardiopulmonary Exercise Physiology in AL Amyloidosis Patients with Cardiac Involvement and Its Association with Cardiac Imaging Parameters"

_jcm, 2022, doi:10.3390/jcm11185437_

Round 1

Reviewer 1 Report

Interesting paper providing a hypothetical new role of CPET in this field. However the small sample size and the lack of a control group severely limit the importance of the data. Can you test if CPET parameters have an additive prognostic role than echo/strain and biomarkers?

Author Response

Reviewer #1:

Interesting paper providing a hypothetical new role of CPET in this field. However the small sample size and the lack of a control group severely limit the importance of the data. Can you test if CPET parameters have an additive prognostic role than echo/strain and biomarkers?

Response:       We appreciate the reviewer’s comment. Indeed the sample size is small but we should take into account the fact that the disease is relatively rare and many of these patients are not capable of exercising at all due to heart failure and impaired functional status. The main strengths are the detailed phenotyping with advanced imaging modalities. Indeed, we performed AUC analysis and did not find an additive prognostic role of these parameters combined likely due to the sample size.

Reviewer 2 Report

General comment:

1.      Citation of references in the text should be changed: the reference number should be placed in angle brackets, and a dot should be placed after the bracket.

S     Specific comments:

Methodology

2.      Although the abbreviation AL CA is listed in the list of abbreviations, it is necessary to explain it when it appears for the first time in the text (line 70).

3.       Check and correct the specified data in “the staging system of Mayo Clinic 2004 based on combinations of NT-proBNP and cardiac TnT at presentation of diagnosis : stage I (NT-proBNP 332 ng/L or high sensitivity cardiac TnT ≥50 ng/L), and stage III (NT-proBNP >332 ng/L and high sensitivity cardiac TnT ≥50 ng/L). Stage IIIB was defined as NTproBNP >8500 ng/L and high sensitivity cardiac TnT ≥50  ng/L” (lines78-83).

4.       If you write MRI, in my opinion it is unnecessary to add the word imaging  (line 90).

Results

5.      This statement should be checked and rewritten : “Per Mayo stage, 4 (17.4%), 11 (47.8%) were stage 2 8(34.8%) stage 3 and per NYHA class at diagnosis was 8 (34.8%) wrer class I, 12 (52.2%) were class II and  3 (13%) class IIIa” (lines  174-176).

6.       Line 206: before the word patients, the letter i is redundant

Author Response

Reviewer #2:

General comment:

  1. Citation of references in the text should be changed: the reference number should be placed in angle brackets, and a dot should be placed after the bracket.

Response: We appreciate the reviewer’s comment. In the revised manuscript we changed reference citation accordingly.

S     Specific comments:

Methodology

  1. Although the abbreviation AL CA is listed in the list of abbreviations, it is necessary to explain it when it appears for the first time in the text (line 70).

Response: We appreciate the reviewer’s comment. We added the term light chain cardiac amyloidosis in line 70 as suggested.

  1. Check and correct the specified data in “the staging system of Mayo Clinic 2004 based on combinations of NT-proBNP and cardiac TnT at presentation of diagnosis : stage I (NT-proBNP 332 ng/L or high sensitivity cardiac TnT ≥50 ng/L), and stage III (NT-proBNP >332 ng/L and high sensitivity cardiac TnT ≥50 ng/L). Stage IIIB was defined as NTproBNP >8500 ng/L and high sensitivity cardiac TnT ≥50  ng/L” (lines78-83).

Response: We appreciate the reviewer’s comment. We corrected all specified data accordingly in lines 78 to 84.

  1. If you write MRI, in my opinion it is unnecessary to add the word imaging  (line 90).

Response: We appreciate the reviewer’s comment. We removed the word imaging from this sentence.

Results

  1. This statement should be checked and rewritten : “Per Mayo stage, 4 (17.4%), 11 (47.8%) were stage 2 8(34.8%) stage 3 and per NYHA class at diagnosis was 8 (34.8%) wrer class I, 12 (52.2%) were class II and  3 (13%) class IIIa” (lines  174-176).

Response: We appreciate the reviewer’s comment. In the revised manuscript we modified the two sentences as follows: According to the Mayo stage classification, 4 patients (17.4%) were at stage I,  11 (47.8%) were at stage 2, and 8 (34.8%) were at stage 3. Regarding NYHA classification at diagnosis, 8 patients (34.8%) were at class I, 12 (52.2%) were at class II and 3 (13%) at class IIIa.

  1. Line 206: before the word patients, the letter iis redundant

Response: We appreciate the reviewer’s comment. Apologies for this mistake, we have amended the sentence accordingly.

Reviewer 3 Report

This research has an original objective and a meaningful content. I congratulate the authors for the work done. I am grateful with the editors for the possibility of revising this manuscript. Although the quality of the manuscript is high, I would like to make some contributions that I hope will increase it and improve readers' understanding. 

Introduction: 

In my opinion,  must be improve. The introduction is clear and well worked, but it is inadequate to reader without formation in this topic. Please, develope it.

References 7 and 8 are missing in text.

Material and methods: 

The study of the design is appropriate and well described, information include in this section allow for replication of this investigation.

Results:

I understand that authors included a list of abbreviations at the end of article, good idea, but is difficult to interpretate some abbreviations in Table 1 and 2. I propose include a footnote in this tables to include this information.

In table 2, please emphasize when differences between groups have statistical significance.

Data analysis: 

The statistical analysis is correct and well described.

Discussion: 

It is well oriented.

Appearance:

My article version in PDF includes a note that minimise text and decrease quality of presentation.

Author Response

Reviewer #3:

This research has an original objective and a meaningful content. I congratulate the authors for the work done. I am grateful with the editors for the possibility of revising this manuscript. Although the quality of the manuscript is high, I would like to make some contributions that I hope will increase it and improve readers' understanding.

Response: We appreciate the reviewer’s comment.

Introduction: In my opinion,  must be improve. The introduction is clear and well worked, but it is inadequate to reader without formation in this topic. Please, develope it.

Response: We appreciate the reviewer’s comment. In the revised manuscript we added the following section to the Introduction, to guide the audience on the potential value of this testing modality in CA: “The rationale behind utilization of cardiopulmonary exercise parameters in cardiac amyloidosis patients is that an important component of functional status limitation is due to cardiac involvement leading to heart failure. Targeted therapies towards the etiology of amyloidosis prevent deterioration of disease but exert minimal effects on the echocardiographic characteristics of the left and the right ventricles. However, many patients treated with disease modifying therapies and supportive care aiming at volume control, experience improvement in functional status. CPET could accurately assess the functional status and the severity of heart failure at baseline before treat-ment, contribute to prognosis and provide valuable information on the disease course and response to treatment. The available data demonstrate severely reduced function-al status on CPET and their population consists mainly of transthyretin patients. In light of the limited currently available evidence focusing on AL amyloidosis, we sought to examine cardiopulmonary exercise testing performance of AL amyloidosis patients with cardiac involvement and identify clinical, biochemical and imaging markers as-sociated with exercise capacity impairment in this challenging patient population.”

References 7 and 8 are missing in text.

Response: We appreciate the reviewer’s comment. This is our mistake and these two references are added to the final large paragraph of the Introduction. The refer to the most recent studies on CPET in cardiac amyloidosis.

Material and methods:

The study of the design is appropriate and well described, information include in this section allow for replication of this investigation.

Response: We appreciate the reviewer’s comment.

Results:

I understand that authors included a list of abbreviations at the end of article, good idea, but is difficult to interpretate some abbreviations in Table 1 and 2. I propose include a footnote in this tables to include this information.

Response: We appreciate the reviewer’s comment. We added the following abbreviations:

“Abbreviations: BMI; body mass index, eGFR; estimated glomerular filtration rate, NT proBNP; N terminal pro brain natriuretic peptide, PETCO2; patient endtidal CO2, RQ; respiratory quotient, VO2; oxygen uptake, VE/VCO2; ventilatory efficiency slope, ECV; extracellular volume, E/Ea; ratio of peak velocity blood flow from left ventricular relaxation in early diastole to peak velocity flow in late diastole caused by atrial contraction, GLS; global longitudinal strain, STDI; S wave at tissue Doppler Imaging of the Right Ventricle, TAPSE; Tricuspid annular plane excursion, IVS; interventricular septum, PW; posterior wall of the left ventricle”

In table 2, please emphasize when differences between groups have statistical significance.

Response: We appreciate the reviewer’s comment. In the revised manuscript we added the symbol “  *   “ in front of statistically significant values.

Data analysis:

The statistical analysis is correct and well described.

Response: We appreciate the reviewer’s comment.

Discussion:

It is well oriented.

Response: We appreciate the reviewer’s comment.

Appearance:

My article version in PDF includes a note that minimise text and decrease quality of presentation.

Response: We appreciate the reviewer’s comment. We apologize for this. However, in our pdf version the quality is appropriate. We will contact the Journal production to amend th